# Bidirectional quantitative scattering microscopy

Kohki Horie[1,3], Keiichiro Toda ®[2,3], Takuma Nakamura ®[2] & Takuro Ideguchi ®[1,2] ✉

Quantitative phase microscopy (QPM) and interferometric scattering (iSCAT) microscopy are powerful label-free imaging techniques that are widely used in biomedical applications. Each method, however, possesses distinct limitations: QPM, which measures forward scattering (FS), excels at imaging microscale structures but struggles with rapidly moving nanoscale objects, whereas iSCAT, based on backward scattering (BS), is highly sensitive to nanoscale dynamics but lacks the ability to comprehensively image microscale structures. Here, we introduce bidirectional quantitative scattering microscopy (BiQSM), an approach that integrates FS and BS detection using off-axis digital holography with bidirectional illumination and spatial-frequency multiplexing. BiQSM achieves spatiotemporal consistency and a dynamic range 14 times wider than QPM, enabling simultaneous imaging of nanoscale and microscale cellular components. We demonstrate BiQSM's ability to reveal spatiotemporal behaviors of intracellular structures and small particles using FS and BS images. Time-lapse imaging of dying cells further highlights BiQSM's potential as a label-free tool for monitoring cellular vital states through structural and motion-related changes. By bridging the strengths of QPM and iSCAT, BiQSM advances quantitative cellular imaging, opening avenues for studying dynamic biological processes.

Quantitative phase microscopy (QPM)[1–5] provides refractive-index (RI) contrasts of specimens with high spatiotemporal resolution, making it a widely utilized technique for label-free live-cell imaging. Unlike fluorescence microscopy, which offers specific molecular information for a limited number of species, QPM can provide more comprehensive information on cellular images, including dry mass distribution[6] and the motion behaviors of diverse cellular organelles[7]. QPM visualizes a map of the optical-phase delay induced by diffraction from objects by imaging the complex optical field, which consists of forward scattered (FS) and unscattered incident light. In principle, FS contains rich information on microscopic structures ranging from the Rayleigh scattering region (<100 nm, referred to as nanoscale in this study) to the Mie scattering region (>100 nm, referred to as microscale in this study), making QPM well-suited for visualizing complex structures,

such as biological cells. However, QPM faces practical challenges in visualizing nanoscale objects within cells due to a limited dynamic range, as these objects move rapidly on the millisecond timescale. Consequently, QPM has been primarily employed for imaging static or slowly changing microscale structures.

On the contrary, backward scattering (BS) detection is recognized for enabling the measurement of rapidly moving nanoscale objects, as BS predominantly contains Rayleigh scattering. Interferometric scattering (iSCAT) microscopy[8,9] is a notable BS-based imaging method that has demonstrated high sensitivity for in vitro imaging of nanoscale objects, such as 2-nm gold nanoparticles[10] and single proteins[11]. Recently, iSCAT has been applied to in vivo imaging of intracellular vesicles[12] and viruses[13], as well as characterizing molecular diffusion dynamics through particle tracking[12] and time-domain frequency

[1]Department of Physics, The University of Tokyo, Tokyo, Japan. [2]Institute for Photon Science and Technology, The University of Tokyo, Tokyo, Japan. [3]These authors contributed equally: Kohki Horie, Keiichiro Toda. ✉e-mail: ideguchi@ipst.s.u-tokyo.ac.jp

analysis[14]. However, BS detection inherently lacks sensitivity to microscale structures due to the limited detectability in the Mie scattering region, resulting in iSCAT's inability to deliver comprehensive quantitative cellular imaging, a capability that QPM offers.

In this work, we address the aforementioned trade-off by integrating the concepts of QPM and iSCAT within a framework of bidirectional quantitative scattering microscopy (BiQSM), where quantitative complex amplitude images of FS and BS are simultaneously captured. Our BiQSM is based on off-axis digital holography (DH) with bidirectional illumination and the spatial-frequency multiplexing method[15] for capturing both FS and BS images simultaneously. This optical configuration enables the use of a single objective lens and an image sensor, ensuring spatiotemporal consistency between the FS and BS complex amplitude images. Our BiQSM achieves a wide dynamic range in scattering imaging, an order of magnitude wider than that of QPM, allowing for the simultaneous visualization of both rapidly moving nanoscale particles and slowly moving microscale structures. Moreover, the concurrent imaging of FS and BS allows for correlation analysis within the microscale Mie scattering region, with each modality capturing distinct and complementary information.

We initially present a 14-fold enhancement in the dynamic range of BiQSM compared to QPM. Furthermore, we underscore the significance of correlation analysis between FS and BS data by imaging nanoparticles in the Mie scattering region, which remain indistinguishable when analyzed using FS alone. Subsequently, we present wide dynamic range live-cell imaging, enabling simultaneous visualization of both the spatial distribution and dynamic behavior of microscale intracellular structures (e.g., lipid droplets) and nanoscale structures (e.g., small particles and cell membrane) via FS and BS

imaging, respectively. Correlation analysis between FS and BS images allows characterization of intracellular particles within the Mie scattering region. Finally, through time-lapse live-cell observation during the cellular dying process, we exhibit the potential of BiQSM as a powerful label-free tool for monitoring cellular vital states through the acquisition of significant spatiotemporal modifications.

## Results

### Principle of BiQSM

The schematic representation of Rayleigh and Mie scattering is illustrated in Fig. 1a. In Rayleigh scattering, which arises from nanoscale objects, both FS and BS waves exhibit constructive interference owing to negligible optical path length differences of the scattered light, resulting in identical FS and BS intensities. Note that nanoscale objects also include thin structures along the axial direction, such as cell membranes, although these structures possess microscale features in the lateral dimension. In contrast, Mie scattering, which arises from microscale objects, involves constructive interference of FS waves from any depth within the sample, whereas BS waves may undergo destructive interference due to varying path lengths from different depths. This results in a pronounced intensity disparity between FS and BS (see Fig. 1a, bottom). In particular, the BS intensity is saturated for objects larger than ~1 μm (near the wavelength of illumination light), leading to information loss of microscale objects.

We define the amplitude ratios between the scattered wave $E_s$ and the incident illumination wave $E_i$, hereafter referred to as scattering-field amplitude (SA), as a parameter to characterize the bidirectionally scattering fields within a unified framework, facilitating direct comparison between FS and BS images. Figure 1b illustrates the phasor

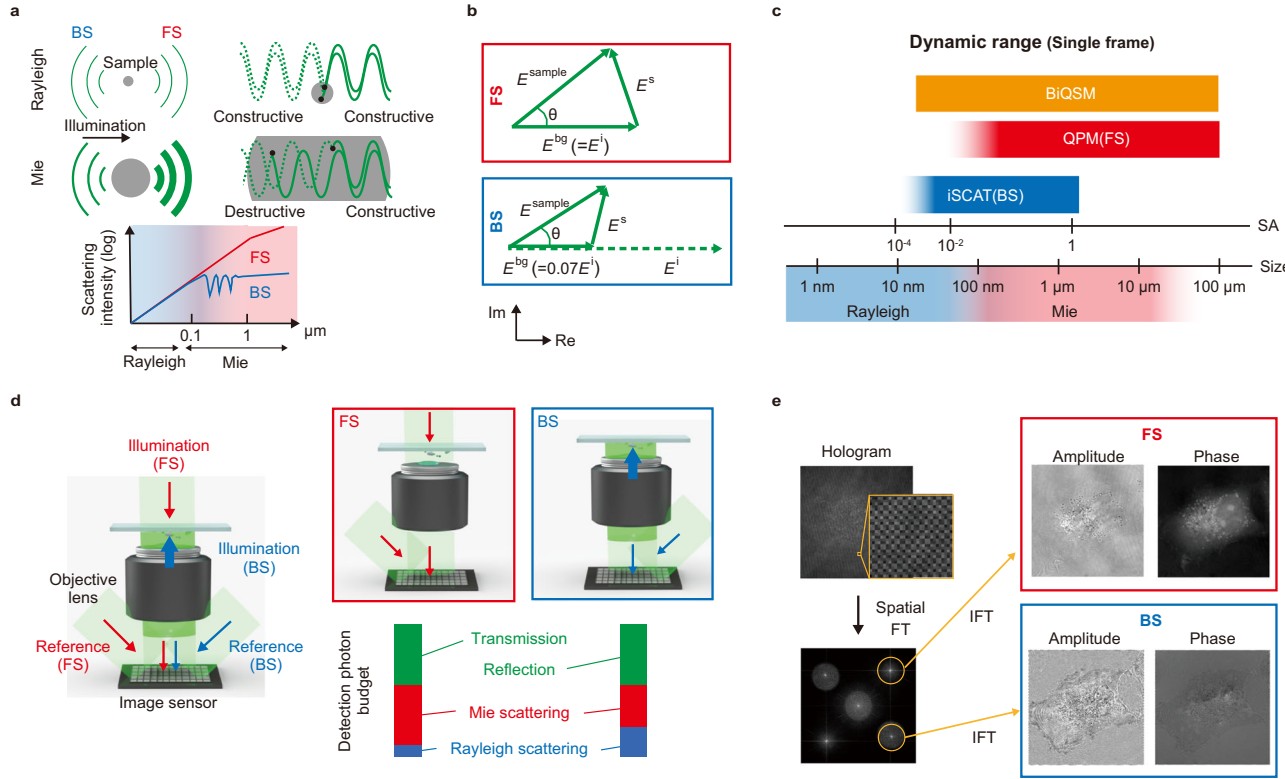

**Fig. 1 | Principle and schematic of the bidirectional quantitative scattering microscopy (BiQSM) system. a** Physical representation of Mie and Rayleigh scattering. **b** Phasor diagrams of the forward scattering (FS) and backward scattering (BS) fields, illustrating the relationship among the incident illumination field ($E_i$), the scattered field ($E_s$), and the measured fields with and without a sample ($E_{sample}$ and $E_{bg}$). **c** Single-shot scattering-field amplitude (SA) dynamic range of BiQSM, quantitative phase microscopy (QPM, a FS technique), and interferometric

scattering microscopy (iSCAT, a BS technique), along with the typical size of scattering objects within biological cells. Note that we assume a QPM technique based on interferometry, such as off-axis digital holography (DH), to calculate the SA dynamic range. **d** Illustrative schematics of FS and BS imaging, along with the photon budget of the image sensor. **e** Reconstruction process of complex amplitudes ($E_{sample}$) for FS and BS measurements from a single hologram.

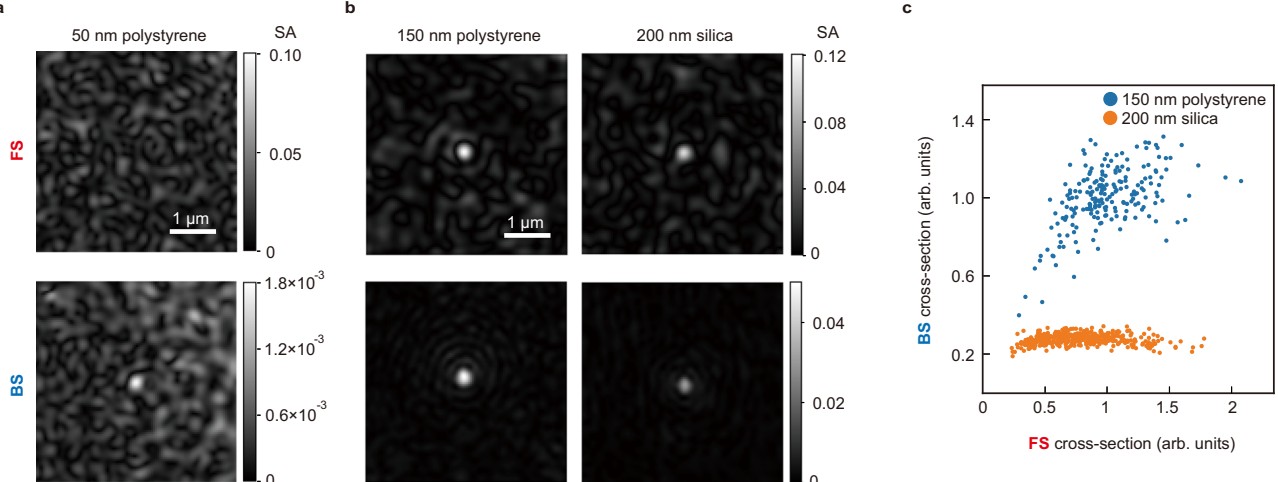

**Fig. 2 | Dynamic range expansion and FS-BS correlation analysis through bidirectional scattering imaging of nanometer-scale beads.** SA images of (**a**) a 50-nm silica bead, and **b** a 150-nm polystyrene bead, and a 200-nm silica bead.

**c** Scattering cross-sections of 150-nm polystyrene beads (151 ± 3 nm, 204 particles) and 200-nm silica beads (203 ± 12 nm, 412 particles) normalized by the average value of 150-nm polystyrene beads.

diagrams of the FS and BS fields, where $E_{sample}$ and $E_{bg}$ represent the measured fields with and without a sample, respectively, following the relation $E_{sample} = E_{bg} + E_s$. Conventional QPM and iSCAT measure $\arg(E_{sample})$ and $|E_{sample}|^2$, respectively. In BiQSM, we derive the SA using the measured complex amplitudes of $E_{sample}$ and $E_{bg}$, as:

$$SA = \frac{|E_s|}{|E_i|} = \frac{|E_{sample} - E_{bg}|}{|E_{bg}/\alpha|} = \alpha \frac{|E_{sample} - E_{bg}|}{|E_{bg}|} \qquad (1)$$

where $\alpha$ is a constant value defined as $E_{bg} = \alpha E_i$, which is the field transmittance and reflectivity of the glass sample holder. The $\alpha$ value is 1 and 0.07 for FS and BS imaging, respectively, representing that fully transmitted light goes into the image sensor in FS imaging, while partially reflected light from the coverslip-water interface reaches the image sensor in BS imaging. The SA value ranges between 0 and 2 by definition. It is important to note that the SA calculation fully utilizes the measured complex amplitudes ($E_{sample}$ and $E_{bg}$), including their phase information. Therefore, computational refocusing can be applied. In this work, we present SA images derived from computationally refocused $E_{sample}$ images. A detailed, comprehensive computational workflow is provided in SA calculation procedure in Methods and in Supplementary Note 2.

Figure 1c illustrates the dynamic ranges of single-frame SA for QPM (FS measurement), iSCAT (BS measurement), and BiQSM (simultaneous FS and BS measurement), all operating under optical shot noise limitations. This figure shows an order-of-magnitude sensitivity enhancement of BS measurement, which is pivotal for the dynamic-range expansion of our BiQSM. Figure 1d represents illustrative schematics of FS and BS imaging, along with the photon budget of the image sensor. In FS imaging, the dominant components are the unscattered, transmitted light ($|E_{bg}|^2 = |E_i|^2$) and the strong Mie scattered light from microscale cellular structures, resulting in a small portion of Rayleigh scattered light from nanoscale objects. In contrast, in BS imaging, fewer photons from the substrate reflection ($|E_{bg}|^2 = 0.07^2|E_i|^2$) reach the image sensor due to the epi-illumination configuration. Therefore, the contribution of Rayleigh-scattered light to the photon budget increases by a factor of 14 (1/0.07) by 204-times stronger illumination without saturating the image sensor. Additionally, BS offers higher sensitivity for Rayleigh scattering because BS from microscale structures is weaker than FS.

BiQSM enables bidirectional scattering imaging with spatiotemporal consistency by simultaneously capturing FS and BS images

using a single detection system. This capability is critical for correlative analysis between FS and BS microscopic images, particularly for samples with densely packed, dynamically varying fine structures, such as cells, where numerical compensation for spatial inconsistencies between different microscopy systems poses significant challenges. To achieve this, bidirectional illumination is applied to the sample from opposite directions, and FS and BS light are simultaneously detected using a single image sensor. The spatial-frequency multiplexing method of off-axis DH[15] is employed to capture both images simultaneously (see Fig. 1d, "BiQSM system" in Methods, and Supplementary Note 1 for details). This method encodes FS and BS information into separate frequency domains by employing distinct off-axis angles for the reference light, enabling selective reconstruction of complex amplitude images ($E_{sample}$) for FS and BS measurements from a single hologram (see Fig. 1e).

### Demonstration of dynamic range expansion and correlation analysis between FS and BS images

To demonstrate the dynamic range expansion with our BiQSM, we evaluated the single-frame minimum detectable SAs for FS and BS imaging by measuring their temporal standard deviations in the absence of samples (see "Temporal noise calculation" in Methods). The evaluated minimum detectable SAs were $6.5 \times 10^{-3}$ and $4.6 \times 10^{-4}$ for FS and BS, respectively, highlighting the superior sensitivity of BS imaging by a factor of 14. These values are consistent with those predicted by optical shot noise, corresponding to scattering signals from 90-nm and 30-nm silica beads ($n = 1.43$) in an aqueous environment.

Then, we measured a 50-nm silica bead to validate this evaluation. Figure 2a presents the FS- and BS-SA images, reconstructed from the measured complex amplitude images using Eq. (1). We imaged a flowing bead to obtain $E_{sample}$ and $E_{bg}$ images from consecutive frames within the same FOV, effectively eliminating temporally static background noise caused by the surface roughness of the glass substrate. In this study, we refer to this approach as temporal differential analysis. As expected from the evaluated minimum detectable SAs, the 50-nm silica bead was detected solely through BS imaging.

Next, we verified the advantage of correlation analysis between FS and BS images using the SA images of a 150-nm polystyrene bead and a 200-nm silica bead, as shown in Fig. 2b (see "Preparation of samples" in Methods for details of the sample). Figure 2c displays the scattering cross-sections of the beads for FS and BS images, derived by spatially integrating the squares of the SA images. The data are normalized by

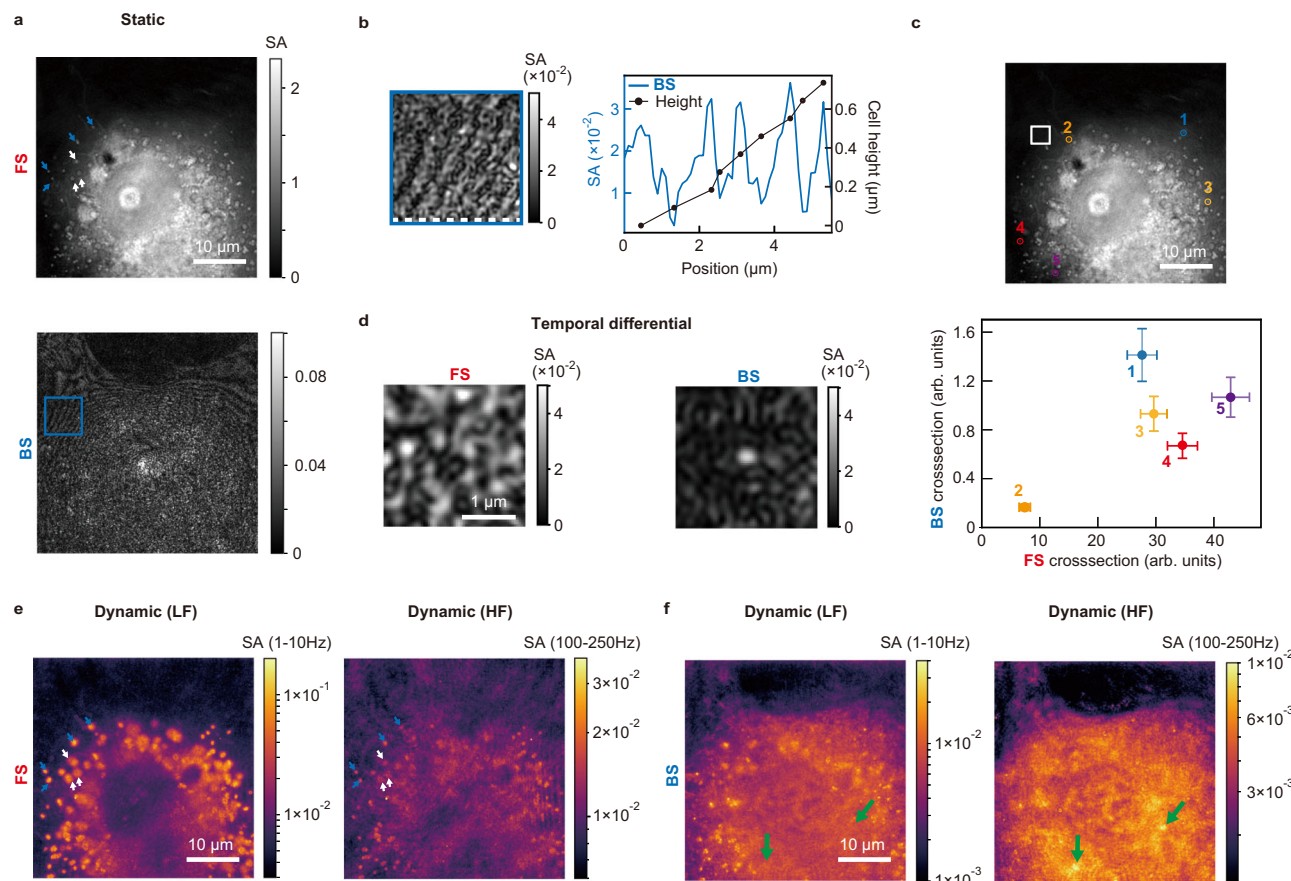

**Fig. 3 | Live-cell imaging with BiQSM. a** Static FS (top) and BS (bottom) images. **b** Left: Magnified image of the blue square region in (**a**), Right: Cross-sectional profile along the white dashed line in the left panel, with estimated cell height. **c** Top: FS image highlighting selected intracellular particles (Particle 1–5) analyzed. Bottom: Scatter plot of FS vs. BS cross-sections for the selected particles. Values are normalized to the average of 150-nm polystyrene beads shown in Fig. 2. The error bars indicate the standard deviation, estimated from the experimental variance of data shown in Fig. 2 (see Supplementary Note 4 for details). **d** Temporal differential FS (left) and BS (right) images of an intracellular small particle. **e** Dynamic low-frequency (LF)-FS (left) and high-frequency (HF)-FS (right) images. **f** Dynamic LF-BS (left) and HF-BS (right) images.

the average cross-section values of the polystyrene beads (see "Beads measurement" in Methods for the calculation procedure). The results show that while the FS cross-sections of the two beads overlap, their BS cross-sections differ, demonstrating that FS-BS correlation analysis effectively differentiates these beads. Note that other sets of beads may exhibit overlapping BS cross-sections but not FS cross-sections, depending on their refractive index (RI) and size combination. FS-BS correlation analysis also provides the ability to decouple RI and size from the SAs, enabling precise determination of bead properties. For instance, the RI and size of the silica beads were evaluated as $1.426 \pm 0.006$ and $206 \pm 18$ nm, respectively, closely aligning with the manufacturer's specifications of 1.43 and $203 \pm 12$ nm. The primary contributors to the measured size variation are the intrinsic size variation of the beads themselves and measurement noise—specifically, surface roughness of the glass coverslip in BS measurements and temporal fluctuations in background FS signals caused by out-of-focus beads in FS measurement (see Supplementary Note 3 for details). This capability of quantitative particle characterization holds significant potential for studying bioparticles, as further explored in the "Discussion" section.

## Bidirectional quantitative scattering imaging of living cells

We applied BiQSM to live-cell imaging. Figure 3a displays FS- and BS-SA images of COS7 cells (hereafter referred to as static images). The static FS image reveals the global structure of a cell and microscale cellular organelles, including the nucleus, nucleoli, and lipid droplets, with SA

values up to ~2. The global SA distribution within the cell closely resembles the phase map obtained via QPM, which is proportional to the spatial distribution of depth-integrated dry mass concentration, a key parameter for monitoring cellular states or growth rates[6]. We validated that the microscale particles detected in FS imaging (e.g., particles indicated by white arrows) are predominantly lipid droplets by molecular bond-specific imaging techniques[16]. In contrast, the static BS image reveals nanoscale structures with SA values an order of magnitude smaller. For instance, as shown in the inset of Fig. 3b, Newton's ring-like interference patterns were observed with an SA of less than 0.01. This interference originates from the reflections on the cell membranes, enabling the evaluation of the cellular height map.

To visualize intracellular particles, we applied temporal differential analysis, as shown in Fig. 2 (see "Intracellular particle measurement" in Methods for the calculation procedure). We first conducted FS-BS correlation analysis on intracellular particles within the Mie scattering region. Figure 3c presents selected particles in the FS image alongside their corresponding scattering cross-sections, derived from FS and BS images. The error bars in Fig. 3c were calculated based on the experimental variance quantified in Fig. 2 (see Supplementary Note 4 for details). The figure clearly demonstrates that the FS-BS correlation analysis enables differentiation of these particles. Further analysis of the obtained data allowed us to extract the RI and size, revealing that these particles are likely small lipid droplets, with typical RI values ranging from 1.44 to 1.50[17,18] (See Supplementary Note 4 for

details). Variations in the measured RI and size may reflect differences in the chemical composition and physiological state of the lipid droplets[18].

Figure 3d displays magnified FS and BS temporal differential images, highlighting an even smaller particle—at or below the spatial resolution—detected exclusively in the BS image. The observed SA value of ~0.04 suggests that it is also likely a small lipid droplet with an RI of ~1.45, rather than an intracellular vesicle, which typically exhibits a lower RI (~1.37)[19]. Investigating such small lipid droplets is crucial for understanding the dynamics of lipid metabolism[20]. While BS imaging enables highly sensitive detection of nanoscale particles, FS imaging suffers from background signal originating not from instrumental noise but from motion-induced fluctuations of microscale cellular structures. These fluctuations obscure the visualization of small particles below ~210 nm with an RI of 1.43, which is evaluated by the standard deviation of the background signal within a particle-free region (white square in Fig. 3c). This limitation may be overcome by implementing modified illumination schemes (see "Discussion" for details).

Although the static and temporal differential SA images have already demonstrated the advantages of this microscope, dynamic image analyses provide even greater insights, particularly for analyzing living cells with rapidly moving particles within slowly drifting larger structures. To visualize these dynamic variations, we continuously captured SA images at 500 fps over 10 s, generating a time-series dataset of 5,000 frames. We then conducted time-domain frequency analysis[21] to map the temporal SA fluctuations of each pixel within two specific frequency bands: 1-10 Hz (hereafter referred to as low frequency, LF) and 100–250 Hz (high frequency, HF) (see "Dynamic image calculation" in Methods and Supplementary Note 5). The LF range was selected to maximize contrast from microscale structures while minimizing the influence of slowly varying background noise caused by air fluctuations and sample stage drift. This range also corresponds to the timescale of characteristic intracellular biomolecular movements driven by bioactivity, distinct from Brownian motion[22]. The HF range was chosen to enhance contrast relative to the LF image. Similar to temporal differential analysis, this approach also visualizes smaller SA values than background static noise caused by the surface roughness of the glass substrate, thereby fully leveraging the wide dynamic range capability of BiQSM.

Figure 3e, f show dynamic LF and HF images for both FS and BS imaging. The signal intensity reflects the object's SA and the strength of its movement within the specified frequency ranges, while the spatial broadening of the signal indicates the object's traveling area. We first discuss the FS images (Fig. 3e). The dynamic LF- and HF-FS images reveal the motion of large lipid droplets, which are also identified in the static FS image. In the dynamic images, droplets indicated by white arrows exhibit spatially broadened signals compared to those indicated by blue arrows, whereas these signals are indistinguishable in the static image. Moreover, the HF-FS signals for these droplets are significantly weaker than the LF-FS signals. These observations suggest that the microscale droplets (white arrows) undergo slower, longer-range movement compared to those in the surrounding region (blue arrows), highlighting the heterogeneity of intracellular fluidity.

Next, we discuss the BS images (Fig. 3f). The dynamic LF- and HF-BS images show evident signals in the nucleus, whereas both FS images display signal intensities comparable to the noise level observed outside the cell. Furthermore, the dynamic HF-BS image highlights localized high-contrast signals (indicated by green arrows) that are undetectable in the LF-BS image, potentially attributed to rapidly fluctuating nanoscale structures. These observations suggest that the dynamic BS images provide unique information not captured by their FS counterparts, likely owing to the dynamic motion of nanoscale objects.

## Time-lapse observation of intracellular structures and their dynamic motions in the cellular dying process

To demonstrate the unique capability of dynamic motion measurement with BiQSM, we conducted a time-lapse observation of the cellular dying process, during which various phenomena, including cell contraction, bleb formation, and ATP depletion, could be observed[23]. Figure 4a presents depth-integrated dry mass concentration maps derived from QPM phase images[6] ($\arg(E_{sample})$) captured at 0, 26, and 53 min after the beginning of the measurement under ambient conditions at room temperature, without $CO_2$ regulation. The dry mass image at 53 min reveals microscale structural changes, such as alterations in nuclear shape and bleb formation (indicated by white arrows), suggesting the occurrence of cellular death. Figure 4b shows the temporal variation in total dry mass within the cell, which remains relatively constant throughout the process, suggesting that most intracellular components are retained within the cell.

Figure 4c presents dynamic LF-FS, and dynamic LF- and HF-BS images, along with their temporal variations within representative regions, highlighting significant changes in the mobility of intracellular components. A particle-rich region within the cytoplasm exhibits distinct features in dynamic FS images, while particle-less cytoplasmic and nuclear regions present characteristic features in dynamic BS images. The LF-FS image and its temporal evolution reveal that the motion of microscale particles, such as lipid droplets, is transiently activated between 10 and 26 min (highlighted in green in Fig. 4c, hereafter referred to as Stage 1), and subsequently damped between 26 and 53 min (highlighted in pink in Fig. 4c, hereafter referred to as Stage 2). BS signals also exhibit characteristic behaviors. LF-BS signals, representing slow fluctuations of nanoscale objects, dropped to ~50% in particle-less and nuclear regions during Stage 2, whereas HF-BS signals, which reflect fast fluctuations of nanoscale objects, increased by ~50% during Stage 1. These trends were consistently observed in all four samples examined (see Supplementary Note 6).

It is worth mentioning that intriguing inter-spatial synchronizations between BS and FS signals occur across different locations. The decrease in the LF-BS signal in particle-less and nuclear regions during Stage 2 synchronizes with the LF-FS signal in the particle-rich region. In contrast, the increase in the HF-BS signal during Stage 1 coincides with the activation of microscale particle movement observed in the LF-FS signal at the particle-rich region. These findings suggest that intracellular components of various sizes may undergo distinct yet interconnected phenomena during the process of cellular death.

## Discussion

We discuss the advantages of BiQSM over other state-of-the-art microscopy techniques that aim for sensitivity enhancement, particularly dark-field (DF) microscopy[24], coherent bright-field (COBRI) microscopy[25], and adaptive dynamic range shift (ADRIFT)-QPM[26]. In DF and COBRI microscopy, a spatial filter is introduced in the Fourier plane to cut or attenuate both the unscattered light and the strong Mie-scattered light from microscale structures. While this approach enhances sensitivity to Rayleigh scattering from nanoscale objects, similar to iSCAT, it inevitably leads to the loss of microscale structural information, even in FS measurements. Consequently, these techniques are not well-suited for achieving wide-dynamic-range cellular imaging. In contrast, the ADRIFT approach effectively mitigates this limitation by employing wavefront shaping techniques. However, BiQSM has two notable advantages over ADRIFT-QPM. Firstly, while ADRIFT-QPM enhances the sensitivity of FS imaging, its ability to detect small particles in living cells is constrained by microscale structural dynamics. In contrast, BiQSM integrates BS imaging, enabling the visualization of nanoscale particles within living cells with higher precision. Secondly, BiQSM facilitates FS-BS correlation analysis, providing a more comprehensive assessment of scattering phenomena.

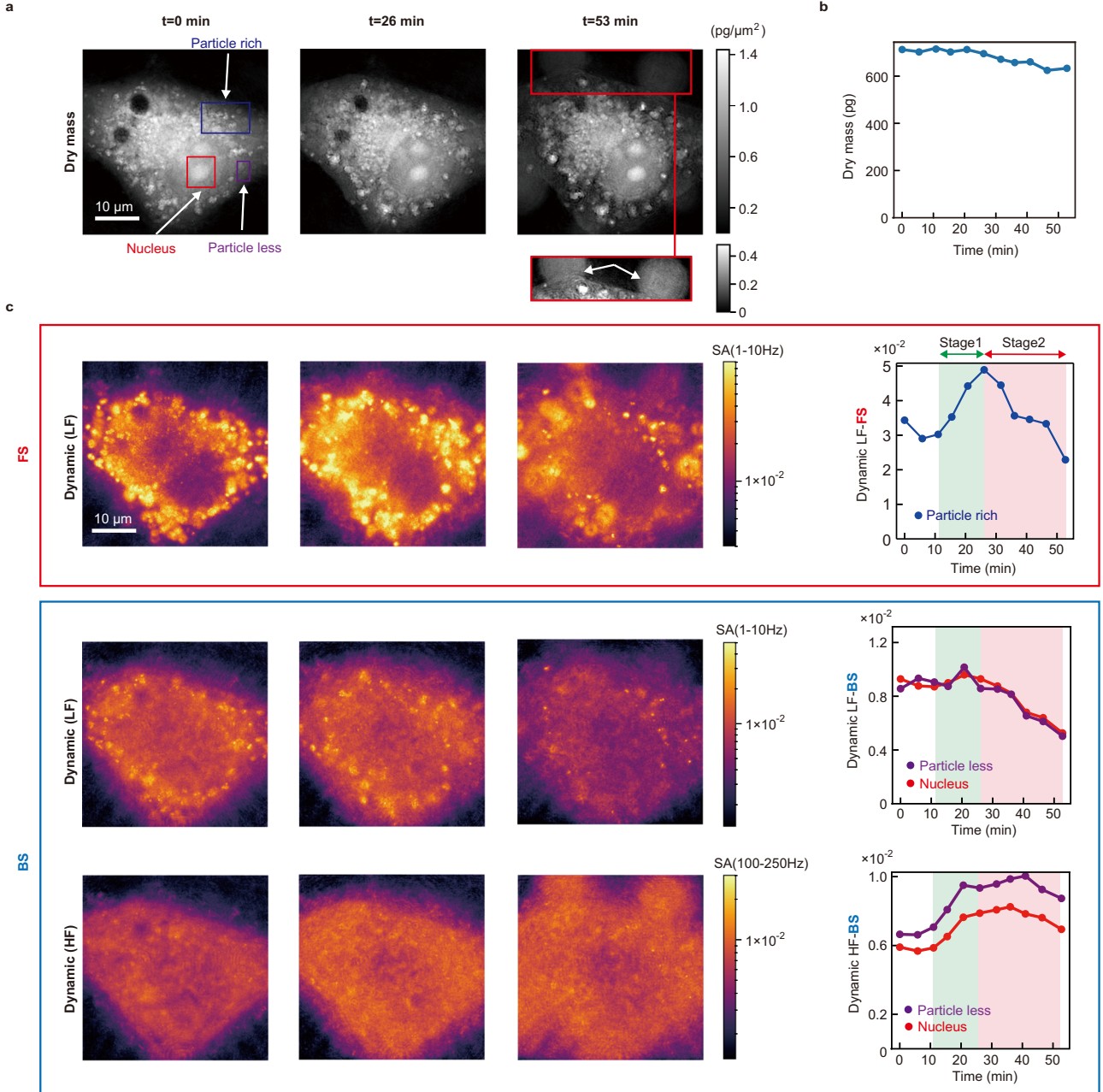

**Fig. 4 | Time-lapse observation of intracellular structures and their dynamic motions in the process of cell death. a** Depth-integrated dry mass concentration maps derived from QPM phase images at 0, 26, and 53 min. **b** Temporal variations of the total dry mass within the cell. **c** Dynamic LF-FS, Dynamic LF-BS, and Dynamic HF-BS images at 0, 26, and 53 min, respectively (left), and their temporal evolutions of dynamic signals within the representative regions (right): A particle-rich region within the cytoplasm for dynamic FS images (blue), and less-particle cytoplasmic and nuclear regions (purple and red) for dynamic BS images.

Recently, another study demonstrated a dual-modal FS and BS imaging technique by combining DF digital holography with conventional intensity-based iSCAT, in which each modality uses separate image sensors and distinct illumination wavelengths[27]. This system was specifically designed for in vitro single-nanoparticle sizing, rather than wide-dynamic-range live-cell imaging. In this approach, the DF configuration shifts the FS dynamic range toward the low-scattering nanoscale region, enabling FS-BS correlation analysis for smaller particles with diameters below 340 nm. However, this shift inherently restricts the ability to visualize microscale intracellular structures. In contrast, BiQSM expands the dynamic range for live cell imaging by integrating FS and BS imaging without shifting the FS dynamic range. Additionally, BiQSM acquires both FS and BS signals using a single

image sensor, ensuring high spatial consistency in correlation analysis —an essential attribute for studying dense samples, such as cells. Furthermore, BiQSM provides both FS and BS complex amplitudes, enabling simultaneous and robust computational refocusing. Additionally, measuring scattering-field amplitude removes two major artifacts commonly encountered in conventional iSCAT imaging: (1) interference between the scattered field and the reflection from the cover slip, and (2) phase distortions caused by the Gouy phase shift of the scattered wave.

There is room for technical improvement of BiQSM. Firstly, the dynamic range of BiQSM, currently constrained by optical shot noise, could be further extended by increasing the number of detected photons. For instance, implementing a CMOS sensor with a higher full-

well capacity (e.g., the Q-2HFW, Adimec) can improve sensitivity by a factor of ~6. Secondly, the bidirectional approach can be adapted to high-dynamic-range optical diffraction tomography. In addition to expanding dynamic range, dual-modal acquisition of FS and BS images also improves axial spatial resolution, as BS information compensates for missing spatial-frequency components along the axial direction in FS imaging[28]. Thirdly, chemical contrast can be integrated into BiQSM by combining it with label-free molecular vibrational imaging modalities, such as mid-infrared[29] and Raman microscopy[30]. In particular, integration with mid-infrared photothermal microscopy[16] could be seamlessly accomplished by incorporating mid-infrared illumination into the BiQSM system. Finally, acquiring additional molecular-specific information through fluorescence microscopy could provide valuable insights into the unidentified contrasts observed in this study.

Finally, we discuss future perspectives based on the findings of this study. The quantitative analysis of RI and size using FS and BS information (as demonstrated in Figs. 2 and 3) can be extended to characterize even smaller biological particles with sizes from 100 nm to 1 μm, both inside and outside cells. Although the current BiQSM system, operating under shot-noise-limited conditions, can detect extracellular particles as small as ~100 nm in both FS and BS imaging, intracellular particles smaller than 200 nm remain undetectable in FS imaging due to substantial background signals, as illustrated in Fig. 3c. These background signals likely originate from the dynamic motion of microscale structures spanning multiple depth layers. To address this limitation, adopting a depth-resolved approach, such as three-dimensional BiQSM, could selectively suppress background signals from irrelevant depth layers, thereby enhancing the detectability of target particles. If the background signal can be reduced by approximately fourfold, the BiQSM system would enable label-free, quantitative characterization of intracellular vesicles, such as exosomes[31], as well as the differentiation of various synthetic nanoparticles and viruses[13].

In another direction, we can further investigate the intracellular dynamics in the cellular dying process presented in Fig. 4. During Stage 2, we observed a synchronized attenuation in the motion of both microscale particles (e.g., lipid droplets) and nanoscale objects. A previous study[32] hypothesized that intracellular fluidity, modulated by ATP-driven fluctuations in actin filaments, accelerates the movement of diffusible cellular components. This suggests that the attenuation observed in Stage 2 may indicate a reduction in intracellular fluidity due to ATP depletion. A comprehensive temporal analysis using fluorescence-integrated BiQSM would enable detailed investigations of actin filament dynamics and ATP distribution, providing insights into the intricate dynamics of the living cellular environment. Additionally, the activation of the motion of microscale particles and nanoscale objects observed during Stage 1 may also be explored using fluorescence imaging. A deeper understanding of these underlying mechanisms could establish such motion attenuation or activation events as potential diagnostic indicators for the early detection of cellular death.

## Methods
### BiQSM system
A detailed schematic of the BiQSM system is provided in Supplementary Note 1. Figure S1 illustrates the visible light source, which is based on the second harmonic generation of a homemade femtosecond ytterbium-doped fiber mode-locked laser at a repetition rate of 63 MHz. After amplification using a ytterbium-doped fiber amplifier and compression with a grating pair (T-1000-1040-12.3 × 12.3-94, LightSmyth Technologies), the pulses are focused onto a periodically poled stoichiometric lithium tantalate crystal (Fan-out PPMgSLT, OXIDE). This setup generates a visible light source at 515 nm with an average power of ~100 mW. The spectral bandwidth is 2 nm, determined by the phase-matching condition. The resulting coherence length of ~709 μm effectively suppresses interference between

scattered light and undesired reflections from optical components, such as internal reflections from the objective lens.

Figure S2 illustrates the BiQSM system, which employs a spatial-frequency multiplexing method for off-axis DH using Mach-Zehnder interferometers. The visible light is introduced into the Mach-Zehnder interferometers in FS and BS imaging systems through a single-mode fiber and a fiber beamsplitter. In the sample arm of the interferometer, the illumination angle is slightly tilted (NA ~ 0.3) only for BS, achieved through two wedge prisms (PS810-A). This tilt is designed to minimize the internal reflected light reaching the image sensor. For both the FS and BS systems, the sample image is magnified in the image sensor plane for holographic measurement (VLXT-17M.I, Baumer) by a factor of 208 using an objective lens (UPLAPO100XOHR, Olympus) and relay lenses. The reference lights are directed to the image sensor at distinct off-axis angles between the FS and BS systems, following adjustments to the optical path length, beam size, and polarization to match those of the sample light. The field-of-view (FOV) and half-pitch resolution are 44 μm and 193 nm, respectively.

### SA calculation procedure
Figure S3 in Supplementary Note 2 illustrates the comprehensive workflow for generating SA images. The recorded hologram is first Fourier transformed into the spatial-frequency domain, where the FS and BS components are separated. Each spectral lobe is extracted using a window defined by the NA (1.33 in an aqueous medium) of the objective lens and then subjected to an inverse Fourier transform to retrieve the complex amplitude $E_{sample}$. The in-focus images of $E_{sample}$ for both modalities (FS and BS) are reconstructed using the angular-spectrum propagation algorithm. A corresponding background measurement, acquired without the sample, is processed using the same procedure to obtain $E_{bg}$. The normalized complex amplitude is then calculated as $\frac{E_{sample}}{E_{bg}} = \frac{E_s}{E_{bg}} + 1$. Next, slight phase drifts in $E_{bg}$ between the sample and background acquisitions are numerically corrected. These drifts predominantly consist of (1) a global, site-independent phase offset and (2) a linear-phase gradient, arising from path-length fluctuations and incident-angle variations between the sample and reference arms of the Mach–Zehnder interferometer, respectively. Both components are removed by fitting a two-dimensional phase profile to the sample-free regions of the $\frac{E_{sample}}{E_{bg}}$ phase map and subtracting the fitted profile. Finally, the SA is derived as $SA = |\frac{E_s}{E_i}| = \alpha|\frac{E_s}{E_{bg}}|$, where $\alpha$ denotes the field transmittance (FS) or reflectivity (BS) of the glass sample holder.

### Temporal noise calculation
For temporal noise evaluation, we captured 500 frames of no-sample images at 500 fps and calculated frame-to-frame differential SA images for both FS and BS imaging. Subsequently, the temporal standard deviation (STD) of the resulting 499 differential images was calculated at each pixel, generating temporal STD maps. The mean value of 200 pixels × 200 pixels in the temporal STD maps was evaluated as temporal SA noise, which is equivalent to the minimum detectable SA.

The theoretical temporal SA noise due to optical shot noise can be expressed as

$$\delta SA^{shot} = 2\sqrt{\frac{(4-\pi)A_{aperture}}{2\nu^2 N_{electron}A_{sensor}}} \qquad (2)$$

where $N_{electron}$ represents the average number of electrons in the hologram contributing to SA image reconstruction, $A_{sensor}$ and $A_{aperture}$ denote the total and cropped pixel areas in spatial frequency space, respectively, and $\nu$ symbolizes the visibility. Equation (2) is derived from the temporal phase noise in off-axis DH[16], incorporating an additional factor of $\sqrt{(4-\pi)/2}$. This factor arises from error

propagation using Eq. (1) under the assumption that the amplitude noise follows the same Gaussian distribution as phase noise. $N_{electron}$ is calculated from the image sensor's output using a full-well capacity of 100 ke$^-$ and a bit depth of 12. Visibility is determined as the ratio of the AC and DC amplitudes in the hologram, obtained by cropping the off-axis and DC regions in the spatial-frequency domain and applying IFT to each, respectively (see our previous publication[16] for details). The values of $A_{sensor}$ and $A_{aperture}$ are 1,048,576 (1024 × 1024 pixels) and 41,548 (π/4 × 230 × 230 pixels), respectively. By substituting these parameters into Eq. (2), the theoretical temporal noise map is calculated.

### Preparation of samples

The polystyrene beads (151 ± 3 nm, ThermoFisher, 3150A) and silica beads (203 ± 12 nm and 52 ± 3 nm, nanoComposix, SISN200-25M and SISN50-25M) were suspended in water within a glass bottom dish for the experiment presented in Fig. 2.

COS7 cells (JCRB, catalog no. JCRB9127) were cultured on a glass bottom dish with high-glucose Dulbecco's modified eagle medium (DMEM), which contains L-glutamine, phenol red, and HEPES (FUJI-FILM Wako). The medium was supplemented with 10% fetal bovine serum (Cosmo Bio) and 1% penicillin-streptomycin-L-glutamine solution (FUJIFILM Wako) at 37 °C in a 5% $CO_2$ atmosphere. In the experiment represented in Fig. 4, time-lapse imaging was conducted at 5-min intervals, starting 10 min after transferring the sample from the incubator to the measurement setup. The experiment was performed at room temperature (23.5 °C) under $CO_2$-depleted conditions.

### Beads measurement

In the experiment shown in Fig. 2, a series of holograms was recorded at 500 fps.

The $E_{bg}$ images were generated by averaging 200 images in which no beads were present. Specifically, a window of 401 frames was selected−spanning from 200 frames before to 200 frames after the beads image shown in Fig. 2. Within this window, the first 100 frames (i.e., frames −200 to −101 relative to the beads image) and the last 100 frames (i.e., frames +101 to +200) were averaged separately. The two resulting averages were then combined to form the final background image.

The FS and BS cross-sections $\sigma_{FS(BS)}$ were determined by spatially integrating the SA images using the following equation:

$$\sigma_{FS(BS)} = \int\int \left| SA_{FS(BS)} \right|^2 dxdy \qquad (3)$$

Prior to integration, numerical focusing was performed on the beads in the SA images, followed by Gaussian fitting to suppress background noise. Temporal averages of the cross-sections over ~100 frames are plotted in Fig. 2c. Trackpy[33] was used for automatic particle detection.

### Intracellular particle measurement

Temporal differential analysis within cells differs slightly from that applied to bead measurements. The SA of intracellular particles is calculated by

$$SA = \alpha \frac{|E_{particle} - E_{no\_particle}|}{|E_{bg}|} \qquad (4)$$

where $E_{particle}$ and $E_{no\_particle}$ denote sequential frames within the same FOV in the presence and absence of the target particles, respectively, and $E_{bg}$ represents the background field acquired from a cell-free region.

In the FS-BS correlation analysis of intracellular particles (Fig. 3c), temporal differential analysis was applied only to BS images, whereas static FS images (Fig. 3a) were analyzed directly. This difference arises from the absence of suitable frames for acquiring $E_{no\_particle}$ in FS sequential images of less-moving, larger particles compared to Fig. 2. However, temporal differential analysis is unnecessary for these particles, as their FS contrast markedly exceeds temporally static background noise from substrate roughness and other intracellular structures. In contrast, prolonged temporal integration of BS frames cancels out particle contrast and provides $E_{no\_particle}$, because slight axial motion of target particles induces dynamic phase change of the scattered field. Accordingly, the average of 5000 frames over 10 s measurement was employed as $E_{no\_particle}$ in BS imaging. To visualize an even smaller, rapidly moving particle (Fig. 3d), temporal differential analysis was applied to both FS and BS images.

To determine the RI and size, we slightly modified the protocol from that used in bead measurements. BS cross-sections were determined by integrating $|SA|^2$ across the particle's pixel region without Gaussian fitting, as cell-specific BS background fluctuations (e.g., membrane motion) reduce fitting accuracy. RI estimates were constrained to the physiologically plausible range 1.38–1.54.

### Dynamic image calculation

The dynamic images were calculated according to the procedure described below. A corresponding figure illustrating this process is presented in Supplementary Note 5. To suppress spatially uniform frame-by-frame temporal variations caused by laser intensity noise and optical path length difference variation between the sample and reference arms, each amplitude and phase image was normalized by the spatial average of the corresponding frame. Additionally, long-term variations exist in the spatial distribution of the background image ($E_{bg}$), primarily resulting from air fluctuations and sample stage drift. These variations exhibit a spatially low-frequency global pattern, as shown in Fig. S5a, where neighboring pixels display nearly identical background fluctuations. To compensate for these long-term variations, the value of each pixel was corrected by subtracting the mean value calculated from its neighboring 11 × 11 pixels (Fig. S5b), effectively removing spatially global background fluctuations. Then, a numerical band-pass filter was applied to the temporal evolution data of each pixel. Finally, the temporal standard deviation of each pixel was calculated to generate the 2D dynamic image map.

### Statistics and reproducibility

Representative images shown in Fig. 2a, b were consistent across over 100 particles. For Figs. 3a, d, and 4a, experiments were independently repeated at least three times using distinct biological replicates, and the representative images showed consistent features across these replicates.

### Reporting summary

Further information on research design is available in the Nature Portfolio Reporting Summary linked to this article.

## Data availability

The raw microscopy datasets generated in this study have been deposited in Zenodo under accession code https://doi.org/10.5281/zenodo.17096218. All source data underlying the figures in this paper are provided in the Source Data file. Additional data supporting the findings of this study are available from the corresponding author upon reasonable request. Source data are provided with this paper.

## Code availability

The data were acquired using Baumer Camera Explorer v3.3. Custom analysis code to generate all figures have been deposited in the repository: https://github.com/Ideguchi-Lab/Bidirectional-quantitative-scattering-microscopy. The data analysis code was written in Python

3.9.12 and used standard Python packages as well as third-party packages such as trackpy v0.6.1. The full list of used packages is available in pyproject.toml and requirements.txt in the repository.

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

## Acknowledgements

This work was financially supported by Japan Society for the Promotion of Science (23H00273, 25H01386, T.I.), JST FOREST Program (JPMJFR236C, T.I.), Precise Measurement Technology Promotion Foundation (T.I.), and UTEC-UTokyo FSI Research Grant (T.I.), the RIKEN TRIP initiative (T.I.).

## Author contributions

K.T. conceived the concept of the work. K.T. and K.H. designed and developed the BiQSM system. T.N. and K.H. constructed the visible light source. K.H. performed the experiments and analyzed the experimental data. K.H., K.T., and T.I. discussed the interpretation of the results. T.I. supervised the work. K.H., K.T., and T.I. wrote the manuscript with inputs from T.N.

## Competing interests

K.H., K.T., and T.I. are inventors of a filed patent application related to the BiQSM system (optical microscopy, JP 2025-35847). This application covers the methodology reported here. T.N. declares no competing interests.
