## [Transparent Peer Review file · Nature Communications]

Bidirectional quantitative scattering microscopy

Corresponding Author: Professor Takuro Ideguchi

Version 0:

Reviewer comments:

Reviewer #1

(Remarks to the Author)

In the manuscript titled “Bidirectional quantitative scattering microscopy”, the authors propose an approach that collects forward and backward scattering from nano- and micro-scale objects in a single-shot. The authors do so by spatial frequency encoding each of the two modalities in a custom-built off-axis digital holographic microscope. The advantage of such bidirectional microscope is two-fold: 1) expand the dynamic range by about an order of magnitude whilst keeping the high sensitivity of BS detection, 2) provide better particle characterisation by looking at the ratio of BS and FS (analogous to flow cytometry based on light scattering that looks at the ratio of forward to side scattering to identify heterogeneity within sample populations).

The authors support their work by first analysing different nanoparticles of varying size and refractive index, showing how their microscope measures the Rayleigh and Mie regime particles. Up to this point I found the authors manuscript a solid piece of work that appeals to a broad audience and a worthy contribution to many communities using a form of holography to characterise nano-to-macroscale objects. However, to my surprise the authors seem to have ignored all the phase information that off-axis holography provides them with and which makes their microscope distinct from other bidirectional approaches (see ref. 26 of the manuscript). Already in Fig1D I was hoping the authors would not only use the scattering amplitude but also the phase information, especially for analysing living cells, which is one of the main advantages of quantitative phase imaging over other label-free approaches. Also using the phase to extract the dry mass concentration seems to be a more straightforward approach than the one developed by the authors.

In the second half of the manuscript, the authors follow live cell dynamics, using their bidirectional microscope combined with time-domain frequency analysis. Specifically, they distinguish different subcellular features and showcase how certain features evolve during cell death as a proof of principle. This section of the manuscript, although well documented and explained experimentally, I find less convincing in terms of showcasing the strength of their platform, simply because the system they are trying to observe is very complex and the interpretations are hard to verify without orthogonal imaging that delivers specificity (fluorescence, bond-selective imaging). I was somewhat expecting to see how the authors would leverage the bidirectionality to for example distinguish synthetic NPs/viruses trafficking within the cells.

Overall the work by Horie et al, is a tale of two parts, which on the one hand I consider very promising and worth publishing in Nat Comms, but on the other hand I would advise the authors make a major revision to address why they have not used the phase information to their advantage.

Some specific comments.

Regarding the use of computational focussing: Please comment whether there was use of any digital refocussing in the manuscript or whether this can be leveraged to obtain additional information you would not get with other bidirectional microscopes.

Regarding the determination of RI: the authors already determined a LUT in Fig S3 that maps their BS/FS ratio to FS to a specific (RI,size). Thus what is stopping the authors from generating RI maps from the cells using this LUT?

Regarding the use of cells: how many cells were used to confirm the observed trends upon initiation of cell death?

Reviewer #2

(Remarks to the Author)
see pdf

Version 1:

Reviewer comments:

Reviewer #1

(Remarks to the Author)

I thank the authors for updating the manuscript and addressing the concerns raised by both me and the other reviewer. I really appreciate the clarification of the computational workflow, this is very helpful. This is an important work that bridges the field of inline holography, specifically that of interferometric scattering microscopy (iSCAT), with off-axis holography. Importantly it's an approach that potentially allows robust characterisation of nanoparticles in both the Rayleigh and Mie regimes. I very much look forward to future work using this technique, especially in the context of nanoparticles and its combination with mid-IR pulses (bond-selective imaging) to get chemical specificity. For instance, it would be fantastic to confirm whether the observed NPs in the cell are indeed lipid droplets vs other intercellular vesicles. All in all, I believe this manuscript is ready for publication.

Reviewer #2

(Remarks to the Author)

The authors addressed most of the comments and considerably improved the manuscript. The workflow and data analysis is now clear which makes the work more accessible. Regarding the cell analysis, I find the updated version more suitable.

From my point of view, I consider the manuscript sufficiently improved to be published.

Minor comments: #3 #4 #5, I was not worried about Gouy phase shifts in the illumination field but in the scattering signal from sub-diffraction limited objects which results in the same issues. This is a general phenomenon but for interferometric scattering, with widefield (plane wave) illumination, the shift is nicely observed in the seminal work from the Sandoghdar group: <https://arxiv.org/abs/2006.15332> It might be worthwhile cross checking the work with this aspect in mind.

The manuscript “*Bidirectional quantitative scattering microscopy*” by Horie *et al.* presents a holographically-multiplexed dual-angle imaging scheme with the goal being to expand the dynamic range with respect to particle size. The paper presents a challenging experiment that is nicely implemented. The manuscript is overall well written. Improvements related to the description of the data analysis (what is actually being used?) are necessary and I would recommend shifting the analysis focus a bit to avoid over-interpretation of some of the highly complex results related to the cellular measurements, more below. Some minor/major rework of these aspects should be sufficient to make the article publishable in Nat. Commun.

Applied to single nano-objects, I find the technique appealing but I had a hard time following parts of the the discussion of, and conclusions drawn from, the cellular measurements. There are several reasons:

1. The hologram processing is completely unclear and it is hence impossible to understand what the “signals” are the authors discuss. In principle, “FS” and “BS”, e.g. forward and back scattered signals, should be complex numbers but the discussion seems to focus on absolute values of the extracted holograms.
2. If absolute numbers are used for analysis then the signals are highly non-trivial as focus and signal magnitudes are tightly coupled (see BS “iSCAT”: “*The point spread function in interferometric scattering microscopy (iSCAT). I. Aberrations in defocusing and axial localization*”. See FS: “*Transport of intensity equation: a tutorial*”). For example, the statement by the authors “*Note that the static FS signal intensity is proportional to intracellular dry mass*” is incorrect for an intensity measurement where defocus is the major factor determining signal levels: the essence of transport-of-intensity equation based phase recovery. If the authors would focus on complex fields, things would look different.
3. Related to 2. and coming back to the correlation analysis (3f) and the papers outlined above, an explanation for the observations could be defocus: for near zero “static FS” objects are in focus where z-motion induced backscattering amplitudes vary slowly. As defocus increases FS increases and the BS signal moves into a region of stronger z-dependence.

To summarise:

I think the forward scattering fluctuation analysis beyond COBRI is great.

I think the conceptual idea is great and has a lot of potential.

The single particle measurements are convincing.

This is a complex experiment and nicely implemented. I especially liked the unification of all iSCATs and holography into a single package. Something the field needs.

In my opinion, the application to complex cellular systems is unfortunate: interference effects, the coupling of z-position and signal levels, Gouy-phase shifts etc. dramatically complicate the data analysis to a point where it is unclear what is actually observed.

I would recommend: Eliminating the correlation analysis and focusing solely on the strong points of FS vs BS, maybe coupled to particle sizes/or regions of interest within a cell. The probability of misinterpretations of this complex relationship does not justify the gain.

Clarify the data analysis, it is imperative to know what is actually being processed.

To Reviewer 1

We are grateful to the Reviewer for taking the time to review our manuscript and give us his/her valuable comments. Our point-by-point responses to the Reviewer's comments are shown below.

Reviewer 1's comment #1:

In the manuscript titled "Bidirectional quantitative scattering microscopy", the authors propose an approach that collects forward and backward scattering from nano- and micro-scale objects in a single-shot. The authors do so by spatial frequency encoding each of the two modalities in a custom-built off-axis digital holographic microscope. The advantage of such bidirectional microscope is two-fold: 1) expand the dynamic range by about an order of magnitude whilst keeping the high sensitivity of BS detection, 2) provide better particle characterisation by looking at the ratio of BS and FS (analogous to flow cytometry based on light scattering that looks at the ratio of forward to side scattering to identify heterogeneity within sample populations).

The authors support their work by first analysing different nanoparticles of varying size and refractive index, showing how their microscope measures the Rayleigh and Mie regime particles. Up to this point I found the authors manuscript a solid piece of work that appeals to a broad audience and a worthy contribution to many communities using a form of holography to characterise nano-to-macroscale objects. However, to my surprise the authors seem to have ignored all the phase information that off-axis holography provides them with and which makes their microscope distinct from other bidirectional approaches (see ref. 26 of the manuscript). Already in Fig1D I was hoping the authors would not only use the scattering amplitude but also the phase information, especially for analysing living cells, which is one of the main advantages of quantitative phase imaging over other label-free approaches. Also using the phase to extract the dry mass concentration seems to be a more straightforward approach than the one developed by the authors.

Authors' response:

We sincerely thank the reviewer for the positive evaluation of our study. We would like to correct the reviewer's potential misunderstanding regarding the statement that "*the authors seem to have ignored all the phase information that off-axis holography provides them.*" Our method does utilize the phase information. Let us describe the computational workflow of our analysis, where the phase information provided by off-axis holography plays a crucial role.

Initially, the hologram is reconstructed into the complex amplitude, $E_{\text{sample}} = E_s + E_{\text{bg}}$, retaining both its amplitude $|E_{\text{sample}}|$ and phase $\arg(E_{\text{sample}})$. This complex amplitude represents a superposition of the background field E_{bg} and the scattered field E_s , where the latter is the primary parameter of interest in this study. To isolate E_s , we independently acquire a background hologram without the sample, extract E_{bg} , and calculate the scattering amplitude via Eq. 1 ($|E_s| = |E_{\text{sample}} - E_{\text{bg}}|$), for both FS and BS components. We emphasize that this calculation inherently leverages both the amplitude and phase information provided by the hologram. These extracted scattering amplitudes, after being normalized by those of illumination light, represent the BiQSM signals (=SA), which enable quantitative comparisons of FS and BS characteristics through scattering cross-section analysis. Although the phase

of E_s is not explicitly presented in the manuscript due to its limited visual interpretability, it is utilized for digital refocusing. This ensures robust retrieval of in-focus information, even in the presence of axial drift. These procedures, along with the pivotal role of phase information, are now explicitly described in the revised manuscript.

We also appreciate the reviewer's insightful suggestion that "*using the phase to extract the dry mass concentration seems to be a more straightforward approach.*" We agree with the reviewer and have adopted this suggestion. Accordingly, we revised Fig. 4 to present dry mass analysis based on the QPM phase image, rather than the FS static SA image. We would like to note, however, that the static FS SA images, which qualitatively resemble the QPM phase images (see figure below), are still useful, as they enable direct comparisons between FS and BS images. Therefore, we retain the SA images in other figures where dry mass quantification is not the primary focus.

Reviewer 1's comment #2:

In the second half of the manuscript, the authors follow live cell dynamics, using their bidirectional microscope combined with time-domain frequency analysis. Specifically, they distinguish different subcellular features and showcase how certain features evolve during cell death as a proof of principle. This section of the manuscript, although well documented and explained experimentally, I find less convincing in terms of showcasing the strength of their platform, simply because the system they are trying to observe is very complex and the interpretations are hard to verify without orthogonal imaging that delivers specificity (fluorescence, bond-selective imaging). I was somewhat expecting to see how the authors would leverage the bidirectionality to for example distinguish synthetic NPs/viruses trafficking within the cells.

Overall the work by Horie et al, is a tale of two parts, which on the one hand I consider very promising and worth publishing in Nat Comms, but on the other hand I would advise the authors make a major revision to address why they have not used the phase information to their advantage.

Authors' response:

We sincerely thank the reviewer for the positive comment that our work would be suitable for publication in *Nature Communications* following a major revision to appropriately incorporate the use of phase information. As explained in our response to Comment #1, phase information was already utilized in the calculation of SA and in digital refocusing. However, we recognized that this was not clearly communicated in the previous manuscript. We have therefore revised the manuscript to clarify these points. Additionally, we have now incorporated dry mass analysis

based directly on QPM phase information, as shown in the revised Fig. 4. We hope these revisions adequately address the reviewer's concern.

Regarding the reviewer's comment that "*the interpretations are hard to verify without orthogonal imaging that delivers specificity (fluorescence, bond-selective imaging)*," we fully agree with it. At the current stage, our obtained data have inherent limitations for biological interpretation. Accordingly, we have carefully revised the relevant sections of the manuscript to adopt a more cautious and appropriately restrained tone. In particular, to address a similar concern raised by Reviewer 2, we have omitted the correlation analyses for the cell images.

We agree with the reviewer that distinguishing synthetic NPs/viruses represents a highly promising application of BiQSM. However, in our current system, tracking small particles around ~100 nm in size within living cells remains challenging. This difficulty arises primarily from intrinsic fluctuations within living cells, which substantially elevate background noise and limit the minimum detectable particle size in the FS modality. This limitation could be resolved by implementing modifications to the illumination scheme (please see our response to Comment #4 for details). The revised manuscript now includes a quantitative discussion of this limitation, as well as potential future directions for applications such as synthetic NPs/viruses tracking (please see Supplementary Note 4 and Discussion in the revised manuscript). Additionally, we have included alternative quantitative analyses of intracellular particles, particularly those abundant in the 300-500 nm size range (please see Fig. 3 in the revised manuscript).

Reviewer 1's comment #3:

Some specific comments.

Regarding the use of computational focussing: Please comment whether there was use of any digital refocussing in the manuscript or whether this can be leveraged to obtain additional information you would not get with other bidirectional microscopes.

Authors' response:

We thank the reviewer for the question. Yes, we performed digital refocusing for the bidirectional SA images by leveraging the complex-field information uniquely accessible through our approach. This capability represents a key advantage of the BiQSM system, enabling robust and quantitative extraction of bidirectional SA and the corresponding scattering cross-sections, even from defocused samples. We have revised the manuscript to explicitly highlight this important advantage.

Reviewer 1's comment #4:

Regarding the determination of RI: the authors already determined a LUT in Fig S3 that maps their BS/FS ratio to FS to a specific (RI,size). Thus what is stopping the authors from generating RI maps from the cells using this LUT?

Authors' response:

We thank the reviewer for the question. The reason we did not generate RI maps is that a straightforward automatic conversion using the LUT does not work reliably across the entire cellular region, as it is only applicable to spherical objects and not to other intracellular structures. Nevertheless, as the reviewer suggested, this LUT remains valuable for analyzing intracellular particles. Accordingly, we performed an additional analysis on intracellular particles—particularly those abundant in the 300–500 nm size range—and have incorporated the results into the revised manuscript.

It should be noted that the current implementation of BiQSM cannot fully exploit the particle characterization capabilities demonstrated in Fig. 2 when applied to live-cell imaging. This limitation arises from fluctuations induced by cellular activity across all axial positions, which are integrated and superimposed in the resulting two-dimensional SA image, thereby increasing background noise. This effect is particularly pronounced in the FS modality, as represented in Fig. 3c. As a result, the minimum particle size that can be reliably distinguished from background fluctuations is approximately 300 nm, larger than the intrinsic detection limit of FS imaging. This background noise can be significantly reduced by incorporating a depth-resolving strategy, such as a multi-angle illumination, as employed in optical diffraction tomography. While this issue was briefly mentioned in the original manuscript, the revised version provides a more detailed and quantitative discussion, along with future directions for investigating smaller intracellular particles below 150 nm in size, such as viruses and vesicles (please see Supplementary Note 4 and Discussion in the revised manuscript).

Reviewer 1's comment #5:

Regarding the use of cells: how many cells were used to confirm the observed trends upon initiation of cell death?

Authors' response:

We acquired data from four cells under identical experimental conditions, including observation time and temperature. The revised Supplementary Information now provides the corresponding datasets for all four cells. The trend presented in Fig. 4 was consistently reproduced across all examined samples.

To Reviewer 2

We are grateful to the Reviewer for taking the time to review our manuscript and give us his/her valuable comments. Our point-by-point responses to the Reviewer's comments are shown below.

Reviewer 2's comment #1:

The manuscript "Bidirectional quantitative scattering microscopy" by Horie et al. presents a holographically multiplexed dual-angle imaging scheme with the goal being to expand the dynamic range with respect to particle size. The paper presents a challenging experiment that is nicely implemented. The manuscript is overall well written. Improvements related to the description of the data analysis (what is actually being used?) are necessary and I would recommend shifting the analysis focus a bit to avoid over-interpretation of some of the highly complex results related to the cellular measurements, more below. Some minor/major rework of these aspects should be sufficient to make the article publishable in Nat. Commun.

Applied to single nano-objects, I find the technique appealing but I had a hard time following parts of the the discussion of, and conclusions drawn from, the cellular measurements. There are several reasons:

Authors' response:

We appreciate the reviewer's comment that some minor/major revisions would render the manuscript suitable for publication in *Nature Communications*. We also fully agree with the reviewer's suggestions to clarify our data analysis and to avoid overinterpretation of complex cellular results. Accordingly, we have revised the manuscript to reflect these points. Below, we provide a detailed point-by-point response to each of the reviewer's comments.

Reviewer 2's comment #2:

The hologram processing is completely unclear and it is hence impossible to understand what the "signals" are the authors discuss. In principle, "FS" and "BS", e.g. forward and back scattered signals, should be complex numbers but the discussion seems to focus on absolute values of the extracted holograms.

Authors' response:

We thank the reviewer for pointing out the lack of clarity regarding the hologram processing in our method. We agree with the reviewer that our previous manuscript may not have sufficiently explained the overall computational workflow. Below, we provide a detailed explanation of the calculation process.

In our method, we begin by applying Fourier analysis—following the standard off-axis holography procedure—to the acquired hologram in order to reconstruct the complex amplitude $E_{\text{sample}} = E_s + E_{\text{bg}}$ for FS and BS measurements, as illustrated in Fig. 1e. The reconstructed field E_{sample} is a superposition of the background field E_{bg} and the scattered field E_s , the latter being the primary quantity of interest in this study. To isolate E_s , we independently acquire a background hologram without the sample to extract E_{bg} , and calculate the scattering amplitude using Eq. 1 ($|E_s| = |E_{\text{sample}} - E_{\text{bg}}|$), for both FS and BS components. We emphasize that this calculation inherently leverages both the amplitude and phase information retrieved from the hologram. Then, the resulting scattering amplitudes are normalized by those of illumination light, representing the BiQSM signals (i.e., SA), which

enable quantitative comparisons between FS and BS characteristics through scattering cross-section analysis. When needed, digital refocusing of the SA image is performed using the complex field information of E_s , ensuring precise axial reconstruction.

In the revised manuscript, we have expanded the Methods section and the Supplementary Information to provide detailed descriptions of these computational procedures, including conventional off-axis hologram processing and digital refocusing methodologies.

Reviewer 2's comment #3:

If absolute numbers are used for analysis then the signals are highly non-trivial as focus and signal magnitudes are tightly coupled (see BS “iSCAT”: “The point spread function in interferometric scattering microscopy (iSCAT). I. Aberrations in defocusing and axial localization”. See FS: “Transport of intensity equation: a tutorial”). For example, the statement by the authors “Note that the static FS signal intensity is proportional to intracellular dry mass” is incorrect for an intensity measurement where defocus is the major factor determining signal levels: the essence of transport-of-intensity equation based phase recovery. If the authors would focus on complex fields, things would look different.

Authors' response:

We thank the reviewer for the comment. As explained in our response to Comment #2, our method does utilize complex fields and therefore does not suffer from the coupling between focus and signal magnitude. The FS and BS signals in BiQSM (i.e., $SA=|E_s|/|E_i|$) are fundamentally different from the intensity representations used in coherent bright-field and iSCAT microscopy (i.e., $|E_{\text{sample}}|^2$). The SA values are rigorously derived from complex fields (E_{sample}) reconstructed via off-axis holography (see Comment #2 for details).

While it is true that the SA contrast may still be affected by defocusing—as is common in coherent imaging methods—the availability of complex-field information in our approach enables digital refocusing. This capability, which we have already implemented in our previous manuscript, effectively mitigates defocus-induced contrast variations. As demonstrated in the particle characterization results in Fig. 2, this ensures robust and quantitative analysis.

Reviewer 2's comment #4:

Related to 2. and coming back to the correlation analysis (3f) and the papers outlined above, an explanation for the observations could be defocus: for near zero “static FS” objects are in focus where z-motion induced backscattering amplitudes vary slowly. As defocus increases FS increases and the BS signal moves into a region of stronger z-dependence.

Authors' response:

As explained in our responses to Comment #2 and #3, our method does not suffer from the defocusing effect observed in conventional coherent imaging. Furthermore, the reviewer's hypothesis may stem from a misinterpretation of the

relationship between the signal and focal positioning, possibly due to an incomplete explanation of the overall computational workflow in our previous manuscript. Unlike coherent bright-field imaging—which typically yields minimal contrast at focus and enhanced contrast under defocus—the static FS image in our method exhibits maximal contrast when the sample is in focus (see Fig. R1 below). Note that the SA contrast observed in the FS image qualitatively resembles the phase distribution obtained via QPM. Additionally, we have confirmed that the dynamic BS contrast within cells remains largely unaffected by axial displacement (see Fig. R2 below), as dynamic BS signals from all axial layers are integrated into the two-dimensional dynamic BS image.

Fig. R1 Dependence of FS static contrast on focal position. The left panel displays a static FS image. The right panel shows cross-sectional profiles of an intracellular particle (highlighted by the red square in the image) at each focal position, obtained via computational refocusing. The baseline static SA value of ~ 0.6 arises from the global intracellular dry-mass contribution.

Fig. R2 Dependence of BS dynamic contrast on focal position. The left panels display the dynamic LF- and HF-BS images, while the right panels show cross-sectional profiles along the white lines in the images for various focal positions, obtained via computational refocusing.

Reviewer 2's comment #5:

To summarise:

I think the forward scattering fluctuation analysis beyond COBRI is great. I think the conceptual idea is great and has a lot of potential. The single particle measurements are convincing. This is a complex experiment and nicely implemented. I especially liked the unification of all iSCATs and holography into a single package. Something the field needs. In my opinion, the application to complex cellular systems is unfortunate: interference effects, the coupling of z-position and signal levels, Gouy-phase shifts etc. dramatically complicate the data analysis to a point where it is unclear what is actually observed.

Authors' response:

We appreciate the reviewer's very positive comments on our method and share the view that the unification of FS and BS modalities introduces a new direction in the field.

We also thank the reviewer for raising concerns regarding cellular imaging. As the reviewer mentions, cellular imaging often encounters various challenges that can affect image contrast. However, we would like to clarify that our BiQSM method is robust to the specific issues mentioned, namely (1) interference effects, (2) defocus, and (3) Gouy-phase shifts. While these phenomena are known to affect iSCAT images, they do not significantly impact the BS SA images obtained using our approach. First, the extracted SA signal is intrinsically decoupled from interference between scattered light and reflected light from the coverslip, as our method isolates only the scattering component E_s from the total complex field $E_{\text{sample}} = E_{\text{bg}} + E_s$ (see our response to Comment #2 for details). Second, the influence of axial displacement on the BS signal amplitude is minimal and does not compromise the integrity of our results, as discussed in our responses to Comments #3 and #4. Finally, in contrast to tightly focused iSCAT systems, our implementation uses plane-wave wide-field illumination, thereby eliminating Gouy-phase shift artifacts in the Gaussian beam. Even if there were a slight Gouy-phase shift in the illumination light, which could influence the phase component of the scattered wave, our SA signal would remain unaffected by such phase variations, as it is represented as an amplitude.

Reviewer 2's comment #6:

I would recommend: Eliminating the correlation analysis and focusing solely on the strong points of FS vs BS, maybe coupled to particle sizes/or regions of interest within a cell. The probably of misinterpretations of this complex relationship does not justify the gain. Clarify the data analysis, it is imperative to know what is actually being processed.

Authors' response:

We thank the reviewer for the two valuable suggestions: (1) to clarify the data analysis, and (2) to consider removing the correlation analysis.

Regarding point (1), we fully agree with the reviewer's suggestion and have revised the manuscript to improve the clarity of the data processing workflow. In the revised manuscript, both the Methods section and the Supplementary Information have been expanded to include detailed descriptions of the computational procedures.

Regarding point (2), we believe that the bidirectional SAs, derived from complex-field information, provide a robust foundation for comparative analyses such as correlation studies, as discussed in our responses to Comments #3, #4, and #5. Nevertheless, we acknowledge the reviewer's concern about the potential for overinterpretation in cellular imaging experiments. In response, we have removed the correlation analysis from our revised manuscript to ensure a more cautious presentation of the data.

To Reviewer 1

We are grateful to the Reviewer for taking the time to review our manuscript and give us his/her valuable comments. Our point-by-point responses to the Reviewer's comments are shown below.

Reviewer 1's comment #1:

I thank the authors for updating the manuscript and addressing the concerns raised by both me and the other reviewer. I really appreciate the clarification of the computational workflow, this is very helpful. This is an important work that bridges the field of inline holography, specifically that of interferometric scattering microscopy (iSCAT), with off-axis holography. Importantly it's an approach that potentially allows robust characterisation of nanoparticles in both the Rayleigh and Mie regimes. I very much look forward to future work using this technique, especially in the context of nanoparticles and its combination with mid-IR pulses (bond-selective imaging) to get chemical specificity. For instance, it would be fantastic to confirm whether the observed NPs in the cell are indeed lipid droplets vs other intercellular vesicles. All in all, I believe this manuscript is ready for publication.

Authors' response:

We sincerely thank the reviewer for the insightful review and for acknowledging that our manuscript has been sufficiently improved and is now suitable for publication.

To Reviewer 2

We are grateful to the Reviewer for taking the time to review our manuscript and give us his/her valuable comments. Our point-by-point responses to the Reviewer's comments are shown below.

Reviewer 2's comment #1:

The authors addressed most of the comments and considerably improved the manuscript. The workflow and data analysis is now clear which makes the work more accessible. Regarding the cell analysis, I find the updated version more suitable.

From my point of view, I consider the manuscript sufficiently improved to be published.

Authors' response:

We sincerely thank the reviewer for the insightful review and for acknowledging that our manuscript has been sufficiently improved and is now suitable for publication.

Reviewer 2's comment #2:

Minor comments: #3 #4 #5, I was not worried about Gouy phase shifts in the illumination field but in the scattering signal from sub-diffraction limited objects which results in the same issues. This is a general phenomenon but for interferometric scattering, with widefield (plane wave) illumination, the shift is nicely observed in the seminal work from the Sandoghdar group: <https://arxiv.org/abs/2006.15332> It might be worthwhile cross checking the work with this aspect in mind.

Authors' response:

We thank the reviewer for the insightful comment. Although we intended to refer to the Gouy phase shifts of the scattering wave in our previous response letter, we mistakenly referred to them as the Gouy phase shifts of the illumination wave. Below is the revised version of our response.

“Our implementation uses plane-wave wide-field illumination, thereby eliminating Gouy-phase shift artifacts in the Gaussian beam. Even if there were a slight Gouy-phase shift in the ~~illumination light~~ **scattering light**, which could influence the phase component of the scattered wave, our SA signal would remain unaffected by such phase variations, as it is represented as an amplitude. “